# Autonomous Vehicle Use and Urban Space Transformation: A Scenario Building and Analysing Method

**Dahlen Silva** *[ID], **Dávid Földes** [ID] and **Csaba Csiszár** [ID]

Department of Transportation Engineering and Economics, Faculty of Transportation Engineering and Vehicle Engineering, Budapest University of Technology and Economics, 1111 Budapest, Hungary; foldes.david@kjk.bme.hu (D.F.); csiszar.csaba@kjk.bme.hu (C.C.)
\* Correspondence: dahlen.silva@kjk.bme.hu

**Abstract:** The use of autonomous vehicles (AVs) has the potential to transform users' behaviour and urban space management. Quantitative and qualitative analyses of impacts require a scenario building method. We considered the fleet size, modal share, car ownership, parking preferences, and urban space repurposing during the elaboration of a novel method. Existing scenarios and results of a questionnaire survey have been used as sources. The method was applied to build scenarios in a case study in Budapest, Hungary. The results were used to calculate the impacts on urban space management, including environmental savings. The key findings are: scenarios with significant shared AV use show that parking demand may be minimised (almost 83%) and urban space repurposing has the highest potential; furthermore, AV use and sharing acceptability may decrease the fleet size and alter the type of shared mode to multiple occupancies. The developed scenario building method serves as a base for future studies. The produced scenarios allow the researchers to focus on the analysis of the impacts caused.

**Keywords:** autonomous vehicles; parking space; scenario building; shared mobility; urban space





## 1. Introduction

Living quality in cities has been decreasing as a consequence of an increase in individual car use, the number of road accidents, traffic congestion, and the time spent driving. Autonomous vehicles (AVs), especially the shared use of them, may mitigate these issues. An AV can drive itself from a starting point to a predetermined destination without human involvement using various technologies and sensors [1].

Increasing the adoption of AVs in shared systems may cause a reduction in the number of vehicles circulating on streets and, consequently, the demand for parking spaces. As a result, it allows changes in the current urban infrastructure, transforming parking spaces to alternative uses such as green spaces, additional traffic lanes, and spaces allocated to cyclists or pedestrians may be possible.

AVs have been considered a disruptive technology with the potential to transform users' behaviour and urban space. Therefore, analysis and assessment of the impacts have become important for understanding how they affect people and living space. Although many studies [2–7] have formulated models and calculations to analyse the impacts of AV use on the number of parking spaces, urban space transformation has not been commonly included in these analyses. Furthermore, guidelines on forecasting future changes in urban space have been scarce.

This research addresses this gap by proposing a scenario building method that supports the assessment of future changes in urban space considering the following group of aspects: acceptability of AVs and shared use, parking preferences, and urban space repurposing.

The novelty of the method is that five aspects (i.e., fleet size, modal share, car ownership, parking preference, and urban space repurpose priorities) affecting the urban space transformation as the consequence of AV use were combined to create scenarios. Moreover,

a combined data source technique was used to build scenarios. The primary data source is a questionnaire survey developed and the secondary data source is a literature review.

With the application of the method, scenarios were created. The scenarios can be used as the basis for detailed analysis concerning the impact of AV use. The created scenarios were applied in Budapest, Hungary, as a case study to evaluate the alteration in the number of parking spaces and the way of repurposing them. A method elaborated previously [8] was used to calculate the space transformation and it was extended to calculate the environmental savings. We found that boosting AVs and sharing acceptability affects fleet size and modal share. As a result, the shared use of autonomous vehicles (SAVs) positively impacts the amount of space for repurposing and air quality.

The method elaborated is complex, and the aim was to lead the readers from the first to the last step. The scenario building method was developed to estimate the future parking demand qualitatively. To determine the alteration in urban space quantitatively, the calculation method for urban space transformation was applied.

The remainder of the paper is structured as follows. In Section 2, the literature is reviewed. The elaborated scenario building method is detailed in Section 3. Moreover, the calculation method for urban space transformation is summarised. The scenarios applied to a case study in Budapest are described in Section 4. Then, the results of the urban space transformation analysis are discussed in Section 5. Conclusions are drawn in Section 6.

## 2. Literature Review

The literature review aimed to identify the most significant aspects that may influence the characteristics of a scenario. The revealed aspects were grouped as:

- Acceptability of AVs and sharing, which influence the willingness to change the car ownership, and furthermore, it may affect the car fleet size and the modal share.
- Parking preference, comparing on-street and off-street parking.
- Possible solutions for urban space repurposing.

The former two aspect-groups impact on parking demand; namely, how many parking spaces could be freed up for urban space repurposing, and possible solutions for urban space repurposing are related to the urban space repurposing priorities. There is a range of possibilities, but the priorities may define which ones will be selected. Urban space repurposing priorities are part of our scenario building method which, consequently, supports scenario building as an output.

### 2.1. Acceptability: AVs and Shared Use

The acceptability is influenced by some factors related to travellers' behaviour: perceived ease of use, attitude, social norm, trust, perceived usefulness, perceived risk, and compatibility. Likewise, factors without behaviour relation are safety, performance-to-price value, mobility (e.g., drunk drivers using AVs), the value of travel time, symbolic value (relation between self-identity and innovation use), and environmentally friendliness [9]. Generally, AV acceptability is related to an individual's background and experiences, which may differ according to regions and countries. For instance, the AVs acceptability is lower (<50%) in countries such as Japan, the Netherlands, and Germany, whereas in developing economies with high levels of traffic congestion and increase in private car ownership, such as the United Arab Emirates, China, and India, the acceptability is high (>70%) [2].

Furthermore, AVs acceptability may also change over time, being influenced by individual preferences, market penetration, government policies, and availability. Therefore, identifying the residents' preferences and needs through a survey may support forecasting the acceptability and impacts of AVs.

Studies focusing on the correlation between demographic characteristics and acceptability of AVs and SAVs have become popular. Even though AVs would be most beneficial for the elder generation, their acceptability level is lower, and the fear of new technology

may be the reason, the so-called technophobia [10–12]. In contrast, students with a high interest in technology would opt for AVs. Besides, elderly travellers require a higher safety level and reliability than younger ones. Consequently, young travellers would be the first users of this technology [11].

Moreover, the willingness to pay for an AV is also higher among younger, highly educated, and higher-income individuals [13]. Significant differences between gender and age are observed, with males and young people having lower reluctance in adopting AVs [14]. Also, a higher willingness to pay was detected among people who have heard about AVs before [13]. Hence, clear information about this technology should be provided in order to overcome the barriers towards acceptability.

Demands towards shared mobility services have been growing, which requires monitoring by the government to assure positive impacts for existing and future uses of public spaces [15]. The main types of shared car ownership are the following:

- Private shared (PSAV): private vehicle usually shared among family members.
- Shared use with single occupancy (SAVSO): use of a vehicle from a car-sharing system by one person on a trip.
- Shared use with multiple occupancies (SAVMO): use of a vehicle from a car-sharing system by more than one person on a trip.

Even though the definition of SAV is mixed in the literature, generally, it refers to SAVSO and SAVMO.

Shared cars might be used for 10% or even 30% of the lifetime [16]. Thereby, several positive impacts can be achieved, for instance, reducing the car fleet size and parking spaces [17]. Besides, SAVs remain parked much less than private vehicles. Also, AVs can move back to the departure point or cheaper parking areas around the city centre [18].

Concerning SAVs acceptability, individuals who never use public transport and drive frequently are less likely to use shared vehicles. On the other hand, individuals with environmental concerns prioritise the shared use [11]. Additionally, the so-called psychological ownership (i.e., the feeling of ownership) was identified as one of the main affecting factors. This feeling can be more important than real ownership and improved with the introduction of customised services (lightning, temperature, music, etc.) [19].

A survey applied in the USA and Israel, countries with a high level of private car ownership, revealed resistance in the acceptability of AVs and shared use: 44% preferred regular cars, 32% would choose a private AV (PAV), and only 24% accept SAVs. Besides, 25% of individuals would refuse the use of SAVs, even if it is entirely free [11]. A survey in Switzerland noted that SAVMO tends to be the most used. Although several forms of AV use were depicted (PAV, autonomous taxis with multiple occupancies, and autonomous public transport shuttles), public transport was the most frequently chosen option [20]. However, a survey in Italy indicated an unwillingness to pay for the new technology. Respondents are willing to pay more (up to €2.31/trip) and travel more (9 min/trip) instead of using SAVs [14].

In addition, another survey in the USA examined the willingness to change a private car in the household to a SAV. Males had a lower willingness to adopt the exchange in single-vehicle households but had a higher willingness in multi-vehicle households. Furthermore, younger and graduated travellers are more likely to accept this change [21]. Another way to address this problem would be the adoption of PSAV. However, its impacts are not significant on reducing the parking demand.

In general, cost, comfort, and time have more impact on the rate of acceptability for long-term mobility decisions than for short-term ones. However, the influence of these aspects on the users' likelihood of purchasing an AV is not significant [20]. Young individuals and individuals with multimodal travel patterns may be more likely to adopt SAVs. Thus, SAVMO is more likely to be used by young travellers, existing car-sharing users, and travellers by car as a passenger. In contrast, travellers by car as a driver are more likely to choose SAVSO. If a respondent travels with public transport, none of the options is more preferred [22].

To sum up, the acceptability of AVs and shared services is influenced by many factors, such as age, feeling of ownership, frequency of car use, and knowledge about the technology. But it can be increased depending on how AVs will fulfil user expectations and will be launched. These two elements are indispensable to be investigated as they affect car ownership (private and shared), fleet size, and modal share, consequently impacting urban space management.

### 2.2. Parking Preference: On-Street or Off-Street Parking

The type and design of residential parking may influence car ownership and car use of households [23]. Also, an increase in car demand from 8% to 14% can be generated depending on whether AVs will be private or shared [24], affecting the number of parking spaces required.

On-street parking occupies considerable street space, which leads to a reduction of the road capacity. Additionally, seeking parking places and the manoeuvring may also reduce the capacity, and even lower road safety and increase the journey time and environmental pollution. Therefore, the number of on-street parking spaces should be mitigated in an efficient transport system [17,25].

To limit the excessive number of parking spaces available in cities, parking policies with a focus on parking maximums have been implemented [26]. However, the success of parking management depends on the residents' acceptability. For example, in Germany, 70% of residents support both demand-oriented policies (e.g., extension of bicycle infrastructure, mobility hubs) and restrictive policy options (e.g., extension of parking fees) [23].

Initially, as a short-term action to mitigate parking spaces, the application of pay and park measures is suggested (e.g., use of parking meters and payment for a limited parking time), mainly in peak hours. As a long-term measure, instead of on-street parking, off-street parking has to be provided near central business district areas within comfortable walking distance (up to a radius of 1 km) [25]. Moreover, mixing different strategies, such as entrance control, capacity restrictions, and different charges among users, would be more effective to reduce parking demand for medium and long terms. Hence, a shift from private to public transport and to other sustainable transport options (e.g., carpooling) may occur [26].

In New Zealand, retailers overestimate how car users are beneficial to their revenue and they consider providing parking spaces as the most important design feature to attract shoppers. On the other hand, shoppers consider pedestrian crossings, wide footpaths, and frequency of public transport services as more important aspects. Also, they are willing to forgo parking in shopping areas. Besides, public, active, and sustainable transport users (representing 40% of the revenue) visit these areas more frequently and spend more time there than car users [27]. For this reason, when available, shopping areas should balance the amount of space dedicated to car users and other consumers in order to attract both. Then, releasing space previously dedicated to car parking to other uses such as bike parking may increase the attractiveness and, consequently, the number of purchases.

A survey in the USA, the United Kingdom, and Germany revealed a 2:1 preference for off-street parking lots or garages over on-street lots if both types were available [28]. However, the choice of a parking space is complex, being influenced by social, economic, and environmental factors. Out-vehicle costs (parking charges, cruising time, and walk time) are more important to users than in-vehicle costs such as fuel cost, travel time, etc. [29,30]. Furthermore, travellers are more sensitive to changes in walking time than parking cost or cruising time [31].

In respect of parking charges, on-street parking should be charged higher than off-street parking, and an hourly fare should be implemented to make on-street parking unattractive, especially for long-term parking [31,32]. For instance, a price increase for on-street parking in a Chinese city (Nanning) reduced parking duration. In contrast, the parking turnover (i.e., the average number of vehicles that are parked within a certain

period) increased after an initial decrease, stabilising in the long-term [33]. Additionally, if parking policies are implemented, users usually shift parking locations rather than switch travel mode. Consequently, a combination of pricing and policy measures should be applied [31]. Applying parking policies that raise parking charges may reduce average car ownership by 17–24% [24].

Measures aiming to reduce cruising and walking time associated with pricing measures showed to be more efficient in shaping parking demand than if they were applied individually [31]. When cruising for an on-street parking space, the uncertainty of finding a spot increases the attractiveness of off-street parking facilities, mainly when the time gained is up to ten minutes [33,34]. However, off-street parking is not the drivers' choice due to the lack of easy access and guidance information, which results in more cruising time and parking on the curb or illegal spaces [32]. Concerning walking time, AV users would not face this problem as AVs would be able to pick them up and drop them off wherever is permitted.

In conclusion, municipalities must interfere in the current situation and develop policies and measures to free up public space from parked vehicles, allowing to repurpose it for a larger community benefit [35]. Understanding the parking demand and preferences, as well as the characteristics of a city and its residents' needs, are valuable features for planning parking policies and urban space repurposing.

### 2.3. Urban Space Repurposing

In many cities of the world, road space is contested, and the allocation of space causes conflicts. Mostly, space is unevenly distributed, and individual motorised transport is more prioritised than sustainable modes such as bicycle. A study in Germany [36] showed that transport space allocation for bicycles is less than 5%. As transport space is limited and area allocation influences the attractiveness of different transport modes and mobility choices, space may have to be allocated differently. Consequently, accidents, air pollution, and noise can be mitigated, and liveability levels can be increased.

The role of individual factors in the overall liveability of a place was investigated via a survey [37]. Three components were analysed: urban form (the built and natural environment along with the infrastructure), urban functions (provided environment for citizens), and urban liveability, which is defined by the quality of the two previous elements expressing how the needs and expectations are met. As a result, urban form and particular mobility-related factors were strongly connected with experiencing a high level of urban liveability.

The repurposing space previously dedicated to car parking and road lanes would allow the creation of new public spaces for sustainable modes of transport, green areas, or wider sidewalks. If the parking space repurpose guarantees an increase in the quality of life, residents are more receptive to demand-oriented on-street parking policies than restrictive ones [23].

Encouraging local assessments to identify residents' preferences about urban green spaces is necessary to meet residents' needs and increase satisfaction. To improve liveability, a study in Portugal [38] revealed that the following actions should be taken:

- Focusing on green space quality rather than quantity in urban planning priorities.
- Investing in small public green spaces rather than in big parks.
- Investing in cleanliness and maintenance within public green spaces and improving plant species richness.

If additional space for a sidewalk is formed from parking spaces, this new public space can be used for cultural activities or leased out for commercial purposes, such as vendors or outdoor seating for restaurants. The revenue generated could be reinvested in further street improvements and public space activation [14]. Therefore, better exploitation of road space might boost the local economy and overall quality of life in an area.

Not only the use of a parking space but also the phases for infrastructure construction and maintenance (materials extraction and transport) add carbon dioxide ($CO_2$), sulphur

dioxide (SO$_2$), and particulate matter emissions [39,40]. With full-scale AV adoption, the footprint of parking may be reduced as AVs could double-park [41] and be smaller due to the average car occupancy below two [42]. These phenomena reduce the amount of space needed in parking areas. Decreasing the demand for parking and fleet size would considerably eliminate part of the emissions. Nevertheless, electric AVs are likely to minimise emissions and maximise the environmental benefits, but it depends on whether electricity comes from renewable energy sources [43] and the emission intensity of the grid is kept at a minimum [44].

Besides the mentioned pollutants, comparing the lifecycles of SAVs and conventional vehicles, SAVs would reduce the energy use, the emission of greenhouse gas (GHG), nitrogen oxide (NOx), volatile organic compounds (VOC), and 10 μm particulate matter emissions (PM$_{10}$), ranging from 5% to 50% [45]. As a result, business models that shift from private ownership to shared-use mobility services have the potential to reduce transportation GHGs [41]. Up to 60% reduction in GHG emissions and up to 94% in CO2 emissions are expected [46]. However, consumer choice represents a crucial element for environmental benefits generated by AVs use becoming reality. For instance, a survey in Germany revealed that private AVs will still be preferred in the future [47].

Moreover, as green spaces are able to remove part of air pollution [48,49], the space re-purposing into green areas can support the mitigation of air pollution caused by road traffic. Besides that, the multifunctional street concept should be embraced by cities, recognising that streets are for mobility, public space, and environmental functions. By balancing these three uses, cities can create urban transformations, mainly improving the quality of the environment and incentivising the use of sustainable forms of mobility [14]. To achieve it, continuous development of city planners' expertise prioritising citizens' broad participation and disciplines in planning is indispensable [50].

Certainly, data collection about the space and people's interactions is essential to justify a project and evaluate its impacts. Useful data may include information on traffic safety, local air pollution, users' satisfaction, and so on [14].

In conclusion, the main findings of the literature review are as follows:

- AVs and shared acceptability are related to many variables influenced by behaviour and demographic characteristics.
- Parking charges, cruising time, and walk time are significant aspects considered in parking choice.
- Citizens desire a high level of liveability through the implementation mainly of green spaces.

## 3. Methods

This section presents the scenario building method, the suggested structure of the questionnaire survey, and the calculation method of urban space transformation in its extended form.

### 3.1. Scenario Building Method

Building scenarios is a way to assess future changes in parking demand and their influence on the transformation of urban space.

The relationship between residents and the urban environment is dynamic and complex. During the scenario building, only the most relevant aspects should be taken into account to simplify the assessment. However, the aspects considered should represent reality. We have revealed the attributes and relationships (Figure 1). The aspects considered are as follows:

- Fleet size: conventional vehicles or AVs.
- Modal share: conventional vehicles, AVs, public transport, active transport, etc.
- Car ownership: private, PSAV, SAVSO, or SAVMO.
- Parking preference: on-street or off-street parking.

- Urban space repurposing priorities: private space, pick-up and drop-off areas, public spaces such as bike lanes, sidewalks, parks, etc.

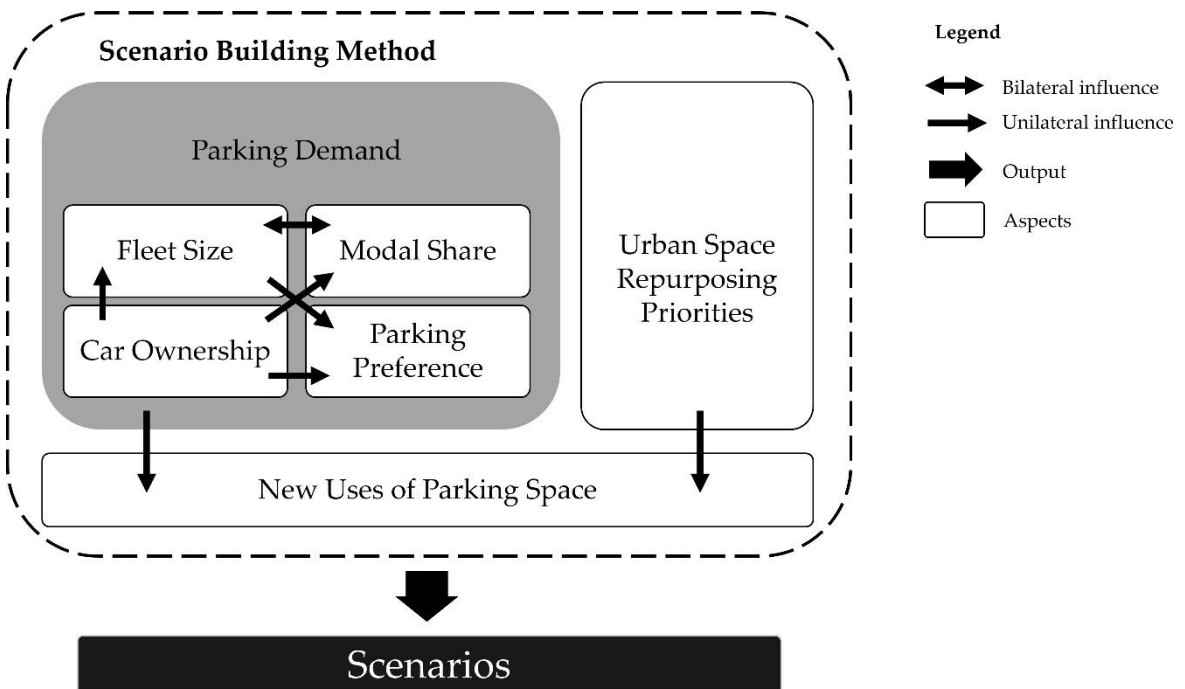

**Figure 1.** Scenario building method: aspects and relationships.

In this paper, the terms fleet size and car ownership refer to the total number of cars and car ownership types (e.g., private or shared), respectively.

According to our scenario building method, parking demand is influenced by fleet size, modal share, car ownership, and parking preference. Fleet size and modal share influence each other (bilateral influence) due to the dynamism of mode choice. The attractiveness of AV-based services distorts modal share. Thus, an oversupply of AVs (i.e., increase in AVs fleet size) may occur to attend to the demand. Consequently, a modal shift from public transport to AVs can happen, which alters modal share and may not contribute to a reduction in parking demand, and shifting from public transport to AV use may increase the parking demand. From this point, it is important that the government interference in creating policies to control the amount of service supply while the quality of public transport is assured to avoid this shift.

Moreover, fleet size influences parking preference. For instance, an increase in fleet size may require the construction of off-street parking spaces because on-street parking spaces already have their capacity overloaded. Also, off-street parking spaces meet the demand for AV use, as AVs can park further from the destination point.

In addition, fleet size, parking preference, and modal share are affected by the car ownership. As sharing increases towards multiple occupancies, fewer vehicles are needed, which results in fleet size reduction. Besides, the use of shared services (SAVSO, SAVMO) may release users from the responsibility of parking choice. SAV operators will be able to choose between on-street parking and off-street parking. According to cost-related aspects, on-street parking is prioritised, though the need for maintenance may request off-street parking. Furthermore, an increase in car ownership indicates a modal shift from public transport or active transport to individual car use. However, the quality of public transport and incentives for active transportation may mitigate this shift.

The variations of these aspects and their combinations influence parking demand. A reduction in parking demand makes parking spaces available for new uses. How the

available space will be used depends on the priorities of urban space repurposing determined by citizens and decision-makers.

The data sources regarding the aspects considered in the scenario building method are presented in Figure 2. Three cases are distinguished:

- Current data are available,
- Current data are not available, and
- Forecast future data to assess future changes.

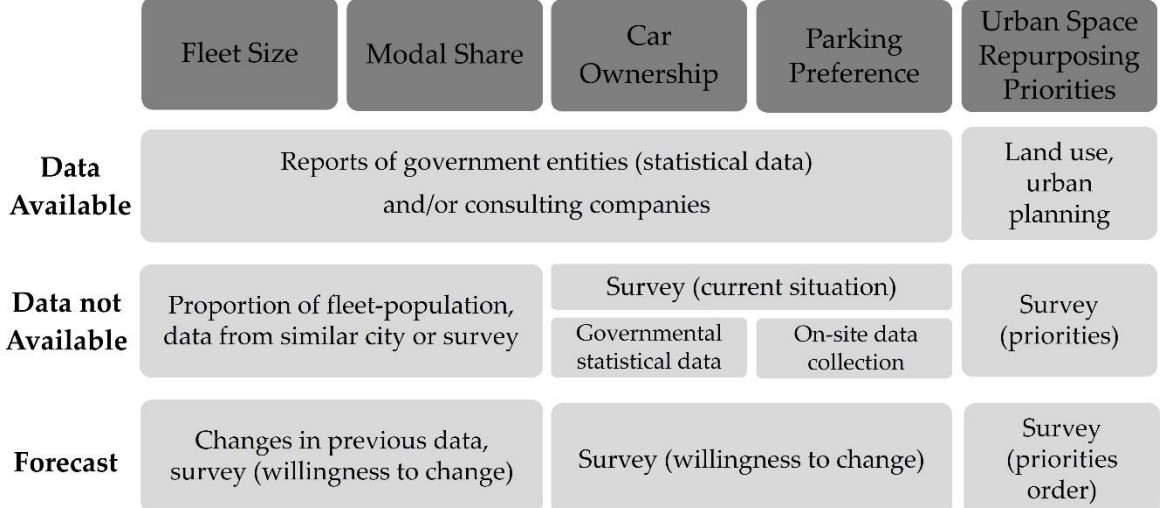

**Figure 2.** Data sources.

For fleet size and modal share, the data can be received from government entities and municipalities, usually from annual reports, and reports of consulting companies that reveal market expectations for the future. Current data support to reveal the current situation, which is sometimes detailed according to the region or city. Comparing previous years' data with the current data contributes to calculating the degree of changes and estimating future situations. However, as AV tends to be a disruptive technology and AV use changes the individual car ownership, it is valuable to investigate the willingness to change and the acceptability of this new technology through a survey, finding out the proportions of conventional and AV fleet and the impact on modal share. If the data are not available or not up-to-date, estimations are needed. For instance, estimations based on the proportion between the size of the fleet in the country and the population or based on data from another city with similar characteristics—similar size, population, and land use.

Data about car ownership and parking preference can be provided by government entities and municipals, mobility service providers (e.g., car-sharing operators), and consulting companies, too. As these data, especially parking preferences, are not usually available, a survey is suggested to gather detailed information, revealing the parking habits in the current situation is possible through on-site data collection. Comparing the available and used parking spaces is an appropriate indicator for expressing preference (demand–supply comparison). Herein, the calculations of the proportion between on-street and off-street parking, occupancy rate, and cost–benefit analysis are feasible.

As urban space is limited, increasing the availability of on-street parking is not an option in many cities. Therefore, for a transitional period, when both conventional vehicles and AVs are present, a slight increase in the fleet may result. Then, a fleet increase may occur if shared mobility services are not able to meet the demand (for instance, long waiting time), thus the use of private vehicles may be more favourable. Conversely, if shared services oversupply the market in order to provide high temporal and spatial availability, the fleet size will be increased. As a consequence, a future increase in fleet size raises

parking demand. Also, the limitation of urban space already reserved for this purpose should be considered during the forecast.

Several data sources can be used to assess future situations. Besides estimation trends from historical data, the expectations and willingness to use different types of transport modes or parking may be collected by surveys. Estimated data and stated preferences are the basis of scenario building. Even though the results from a survey are beneficial in forecasts, they need to be continuously updated as the use and impacts of AVs are not entirely well-known. Namely, people's behaviour and needs may change quickly.

Car ownership is one of the most important aspects influencing the parking demand. As sharing of a vehicle can replace the use of many other private vehicles, the need for parking spaces decreases. For example, if a shared vehicle from a car-sharing system can serve ten users who previously had private cars, a saving of nine parking spaces is resulted. Parking policies can incentivise off-street parking and its acceptance. As a result, urban space would be available for repurposing not only because of the use of shared mobility forms but also due to the change in parking preference to the off-street type.

The future intentions of cities (i.e., priorities) towards urban space repurposing can be collected from reports and policy documents, such as the Sustainable Urban Mobility Plan (SUMP). The individuals' expectations sometimes contradict the collective expectations (for instance, reducing the parking spaces is a collective expectation, but the individual wants to park the car as close to his/her living place as possible). Therefore, the responsibility of urban planners is high.

If data are available and a survey is also possible, it is recommended to use both validation tools. For instance, comparing the results, generating maximum, average, and minimum levels of the forecast, and observing how much they differ.

The questionnaire survey (see Appendix A) has been elaborated to reveal the aspects needed for the scenario building; namely, current situation and change in habits and preferences regarding the car ownership, the acceptability of AVs and shared use (fleet size and modal share can be deduced), parking, as well as to determine the priorities in urban space repurposing. The suggested questionnaire survey contains the following sections:

1. General travel behaviour: the current car ownership and modal share.
2. Parking: most used type of parking.
3. Willingness to change: the willingness to change AVs and shared mobility forms, as well as the situations in which the use of AVs and shared mobility forms would be favourable.
4. Priorities of citizens and decision-makers: priorities about reallocation of urban space.
5. Socioeconomic data: age, gender, income, number of children in the household, education level, and employment status.

In our approach, citizens, experts, and policy makers answer the same questions. Citizens answer the questions according to their best knowledge and information. This knowledge and presumptions can influence the acceptability of AVs.

### 3.2. Calculation Method for Urban Space Transformation

In order to apply the scenarios built and evaluate the impacts of changes in urban space transformation, a calculation method [8] previously developed by the authors was used. Hereby, a summary of the proposed method is given. Furthermore, new Equations (8) and (9) were introduced to calculate the environmental savings generated by the reduction in parking spaces and urban space repurposing options. The future parking demands can be forecasted by the calculation method considering the aspects involved in the scenario building method. Data about the reduction in the number of parking spaces [2–7] and air pollution [39] found by previous studies were used as input, as well as the results of the questionnaire survey. The used indices and variables are summarised in Tables 1 and 2.

**Table 1.** Indices.

| Sign | Name | Set of Value |
|---|---|---|
| k | Parking alignment | k = 1 . . . z (1: parallel, 2: 45°, 3: 90°, etc.) |
| j | Car ownership | j = 1 . . . m (1: private, 2: private shared, 3: shared with single occupancy, 4: shared with multiple occupancies, etc.) |
| x | Type of repurposing | x = 1 . . . y (1: public transport, 2: improving walkability, 3: bike-related strategies, 4: additional traffic lane, etc.) |
| e | Pollutant | e = 1 . . . f (1: carbon monoxide (CO), 2: sulphur dioxide ($SO_2$), 3: nitrogen oxide (NOx), 4: volatile organic compounds (VOC), 5: 10 μm particulate matter emissions ($PM_{10}$)) |

**Table 2.** Variables.

| Sign | Description |
|---|---|
| $VI_j^{CO}(t)$ | rate of increase in the number of conventional vehicles in year t |
| $CO_j(t-1)$ | number of conventional vehicles in the year (t − 1) according to j ownership |
| $VI_j^{AV}(t)$ | rate of increase in the number of autonomous vehicles (AVs) in the year t according to j ownership |
| $AV_j(t-1)$ | number of AVs in the year (t − 1) according to j ownership |
| $SP_j^{AV}$ | percentage of saved parking space caused by autonomous vehicles according to j ownership |
| OFS | percentage of off-street parking space from the total parking spaces |
| $Pe_k$ | percentage of parking spaces according to k alignment |
| $A_k$ | area of each parking space according to k alignment ($m^2$) |
| Rpr(t) | rate indicating how many private areas will be created in year t |
| Pp(t − 1) | percentage of the private area created in the previous year (t − 1) |
| Rpd(t) | rate indicating how many pick-up and/or drop-off areas will be created in year t |
| Pd(t − 1) | percentage of pick-up and drop-off areas created in the previous year (t − 1) |
| $R_x$ | rate of space applied for new use related to x new use |
| $A_x(t-1)$ | area for x new use correspondent to the previous year (t − 1) ($m^2$) |
| F(t) | fleet size in the year t—the sum of conventional and autonomous vehicles |
| $G_{ej}$ | rate of reduction of pollutant *e* according to the car ownership j |
| $G_x$ | rate of the positive impact caused by new use x of urban space |
| $E_e(t)$ | environmental savings on the pollutant e (e.g., ($g/m^2$), (%)) |
| $E_x(t)$ | environmental savings of the new use x (e.g., ($g/m^2$), (%)) |

The steps of the method are as follows (the source of Equations (1)–(7) is our previously developed method [8]):

Step 1: Number of on-street parking spaces according to demand [8]:

$$D(t) = \left[\sum_{j=1}^{m}\sum_{t=1}^{n}\left(1+VI_j^{CO}(t)\right)\times CO_j(t-1)+\sum_{j=1}^{m}\sum_{t=1}^{n}\left(1+VI_j^{AV}(t)\right)\times AV_j(t-1)\times\left(1-SP_j^{AV}\right)\right]\times(1-OFS) \quad (1)$$

The model developed for parking demand considers the number of parking spaces needed every year to accommodate the total fleet. The number of parking spaces for the existing private car and for the possible use of SAVSO services as well as for the other types of ownership are also taken into account.

Step 2: Demand for parking area ($m^2$) [8]:

$$P(t) = D(t) \times \sum_{k=1}^{z}(Pe_k \times A_k) \quad (2)$$

Step 3: Percentage of saved spaces used for private areas, pick-up, and drop-off areas [8]:

$$Dm(t) = Pp(t) + Pd(t) \quad (3)$$

$$Pp(t) = (1+Rpr(t))\times Pp(t-1) \quad (4)$$

$$Pd(t) = (1+Rpd(t))\times Pd(t-1) \quad (5)$$

Step 4: Area destined for new purposes (m$^2$):

$$N(t) = (P(t - 1) - P(t)) \times (1 - Dm(t)) \tag{6}$$

$$N(t) = \sum_{x=1}^{y} (1 + R_x) \times A_x(t - 1) \tag{7}$$

Step 5: Percentage of environmental savings:

$$E_e(t) = \sum_{j=1}^{m} \left( \frac{AV_j(t)}{F(t)} \times G_{ej} \right) \tag{8}$$

$$E_x(t) = A_x(t) \times G_x \tag{9}$$

## 4. Results

In this section, the scenarios created are presented as well as the results concerning the assessment of future changes in a case study in Budapest, Hungary.

### 4.1. Scenarios

The answers from the questionnaire survey were considered during the scenario building. The questionnaire was created in Google Forms in Hungarian and English and answered by 150 respondents of Budapest, Hungary. The answers were used to confirm or disprove the result of the literature review. Although the number of respondents is not high, initial consequences could be drawn.

Answers from citizens and relevant actors from the transport field, such as operators and experts, were gathered anonymously. Also, five policy makers were asked to fill out the questionnaire during private conversations. Though the method does not distinguish the origin of the answers, the variety of respondents was important to ensure reliability.

Most of the respondents were male (67%), from 18 to 35 years old (90%), students (64%), living in a household with more than 2 adults (43%), without children in the household (77%), the master level of education (41%), and possessing a driving licence (78%).

Regarding the acceptability of using AVs, very likely and likely sum up 52%, unlikely and very unlikely sum 37%, and the rejection is 11%. The rejection of sharing an AV with someone during a trip was 9% (SAVMO). The most probable situations to occur for sharing were: 23% would share an AV when going to work, 18% when riding alone, and 13% at times of the day that the respondent thinks it is safe.

Five scenarios describing the current, transitional, and future periods, combining different types of car ownership and fleet size, were created to quantify the gradual changes. The aim was to analyse whether AV acceptability and adoption of shared vehicle use bring significant benefits to reduce parking demand and, consequently, the possibility to repurpose urban space. The scenarios were created according to the result of the literature review and the questionnaire survey. A summary of the scenarios is presented in Table 3.

Scenario 1 (Current) and Scenario 2 (Near Future) represent the current type of ownership and the willingness to change respectively, revealed by the questionnaire survey. The only difference between the scenarios is the rate of different types of vehicle use.

Scenario 3 (Transitional) contains changes in the characteristics of Scenario 2. A mix of conventional and AVs fleet is based on the acceptance of AVs use revealed by the questionnaire survey as well. The percentage of rejection to use AVs (48%) reflects the percentage of using a conventional fleet. Moreover, using a shared AV with single-occupancy service is around 16%, resulting from the sum of 7% shared single-occupancy use from Scenario 2 with the 9% rejection to use a shared service with multiple occupancies. Results of private and private shared ownerships were proportionally distributed based on Scenario 2 proportions.

Scenario 4 (Future with AVs) contains changes in Scenario 3, considering the result of the survey and forecasts from scientific research and experts. The questionnaire revealed

rejection of 11% regarding the use of AVs, resulting in 89% of a fleet composed of AVs. Furthermore, for optimistic scenarios, AVs will be around 75% of the total fleet by 2040 [51], 87% by 2045 [52], and could reach almost 90% in 2050 [53,54]. Therefore, the AVs fleet was considered an average of 88% (between 87% and 89%) that can be reached by 2045. Also, considering that people with previously rejecting attitudes to use AVs and shared services would change, 36% shared use was added to SAVSO.

**Table 3.** Scenarios.

| Scenario | Name | Fleet Type | Details ($CO_j(t-1)$, $AV_j(t-1)$) |
|---|---|---|---|
| 1 | Current | Conventional Vehicles | 21% private; 44% private shared autonomous vehicle (PSAV); 13% shared autonomous vehicle with single occupancy (SAVSO); 22% shared autonomous vehicle with multiple occupancies (SAVMO) |
| 2 | Near Future | Conventional Vehicles | 43% private; 31% private shared; 7% SAVSO; 19% SAVMO |
| 3 | Transitional | Conventional Vehicles AVs | 48% (private and/or shared) 10% private; 7% PSAV; 16% SAVSO; 19% SAVMO |
| 4 | Future with AVs | Conventional Vehicles AVs | 12% (private and/or PSAV) 10% private; 7% PSAV; 52% SAVSO; 19% SAVMO |
| 5 | Mix of SAV fleets | AVs | 10% private; 50% SAVSO; 40% SAVMO |

Scenario 5 assesses a situation in which only AVs are used with mixed types of ownership. It was built for measuring the impact of an expressive shared fleet with multiple occupancies. AVs fleet will be nearly 100% of the car fleet by 2050 [55], having a SAV fleet of over 90% [56]. As a result, Scenario 5 was modelled with 40% SAVMO [6] because a significant rise in the use of shared multiple occupancies is expected. The private proportion remained the same as in Scenario 4. The rate of shared service with single occupancy is 50% based on Reference [5]. It was assumed that population and land use would remain the same.

The type of repurposing the saved space ($x = 1 \ldots y$) is determined according to the priorities of respondents given by the survey results. According to their priority, respondents ranked the new uses from 1 (highest priority) to 6 (lowest priority). For example, as the highest priority, 43% of respondents prefer to repurpose the saved space with green spaces, 15% with bike lanes, 14% with new buildings, 12% with additional traffic lane, 10% with multifunctional zones, and 6% with pick-up and drop-off areas for shared mobility services or individual cars. As second priority and so on, these percentages change.

The percentages regarding new buildings ($Pp(t - 1)$) and pick-up and drop-off areas ($Pd(t - 1)$) in the previous year ($t - 1$) serve as input data in Equations (3)–(5). Considering changes in urban space year by year, planners can use the results from the questionnaire directly (Equation (3)), applying the percentages of the highest priority for the first year, the percentages for the second-highest priority for the second year, and so on. However, planners may establish a rate of changes for the next years, generating the inputs $Rpr(t)$ and $Rpd(t)$, and calculating $Pp(t)$ and $Pd(t)$ differently (according to Equations (4) and (5)). For instance, planners can estimate a reduction of 10% in repurposing for private buildings and an increase of 15% in repurposing for pick-up and drop-off areas for the second year. As a result, using Equation (4), $Rpr(t)$ equals –0.1 and $Pp(t)$ equals 12.6%; in addition, using Equation (5), $Rpd(t)$ equals 0.15 and $Pd(t)$ equals 6.9%. For other types of new uses (e.g., green spaces), the same logic is applied in Equation (7).

Additionally, considering that 50% of parking demand would migrate from on-street parking to off-street parking (OFS), all scenarios were assessed to observe the gains brought by this solution.

In order to quantify the changes during the years of the forecast, the data from scenarios regarding the fleet type ($CO_j(t - 1)$, $AV_j(t - 1)$) were served as input as well as the result of previous studies regarding the reduction in the number of parking spaces ($SP_j^{AV}$) and air pollution ($G_{ej}$, $G_x$) caused by SAVs. The used data and sources are shown

in Table 4. In summary, from 9.5% to 94.4% reduction in parking spaces is observed by existing studies.

**Table 4.** Studies about autonomous vehicles (AVs) use and reduction in parking spaces.

| Ownership | Study | Characteristics | Reduction in Parking Spaces |
|---|---|---|---|
| PSAV | [3] | SAVs among the members of the same household | −9.5% to −12.3% number of private vehicles |
| SAVSO | [2] | 33% or 100% SAVs | −16% or −48% parking spaces |
| | [4] | 5% market SAV | −90.3% and −92.4% parking spaces |
| | [5] | High occupancy of public transport (PuT); 100% or 50% SAVs | −89.3% or −21.2% parking spaces |
| SAVMO | [6] | Autonomous taxi fleet; 0%, 20%, and 40% of penetration rate | 3 conventional cars could be replaced with 1 autonomous taxi with 40% penetration rate |
| | [7] | 2% of market penetration level | −90% parking spaces |
| | [5] | HHigh-occupancy of PuT; 100% or 50% SAVs | −94.4% or −24.2% parking spaces |

Regarding environmental savings, the total energy use can be reduced by 12% with the use of SAVs. Also, the emissions can be reduced by the following percentages [36]: 12% GHG, 20% $SO_2$, 20% NOx, 35% CO, 50% VOC, and 7% $PM_{10}$. Moreover, each square metre of green space with trees or shrubs can annually remove the quantity of pollutants [46,49]: 0.4 g $SO_2$, 0.4 g CO, 0.6 g NOx, 5.8 g $PM_{10.}$

*4.2. Urban Space Transformation: Case Study in Budapest*

The area of study (approximately 673,220 $m^2$) is an old area of the city of Budapest, Hungary, around 120–150 years old, with most of the buildings designed without garages. To attend the demand for parking spaces, on-street parking spaces were created, mainly taking part of the sidewalks that were rather large before this change. This appropriation of part of the sidewalk is visible on-site, where delimitation of parking spaces is painted on them and signs indicate if the car should be parked totally or partially on the sidewalk. A total of 2496 on-street parking spaces were counted during the on-site visit via manual data collection. Cars parked were counted and free spots measured to estimate the number of cars that could be parked in the study area.

The scenarios created were applied to the area of study in order to analyse the possible impacts of AVs on urban space transformation. The developed and extended calculation method was used. The main findings derived from the application of the calculation method are as follows:

- Scenarios 1 (Current) and 2 (Near Future): saved space does not result, the number of parking spaces needed is the same as today.
- Scenario 3 (Transitional): one-third of the on-street parking spaces (818 units) can be eliminated.
- Scenario 4 (Future with AVs): double the space saved in Scenario 3 can be eliminated (65%, 1618 parking spaces).
- Scenario 5 (Mix of SAV fleets): almost 83% of parking spaces (2072 units) can be eliminated.
- Increasing the SAV fleet leads to a reduction in pollutants compared to conventional vehicles. Gains in the reduction of pollutants varied from 2% to 18% (Scenario 3: Transitional) and increased proportionally to the increase in SAV fleet, reaching 5% to 45% (Scenario 5: Mix of SAV fleets).
- If green spaces are formed in the saved space, the current air pollution level can be mitigated. Shrubs planted in all the saved space might remove up to 13 kg of $SO_2$ and CO, 19.4 kg of $NO_x$, and almost 188 kg of $PM_{10}$ annually in Scenario 5.

Figure 3 presents the number of parking spaces needed and space saved in $m^2$ in each scenario. The reduction of on-street parking spaces required can be further improved if the

demand remained will be served by off-street parking places. The figure also includes the amount of space saved if 50% of parking spaces are created in off-street parking.

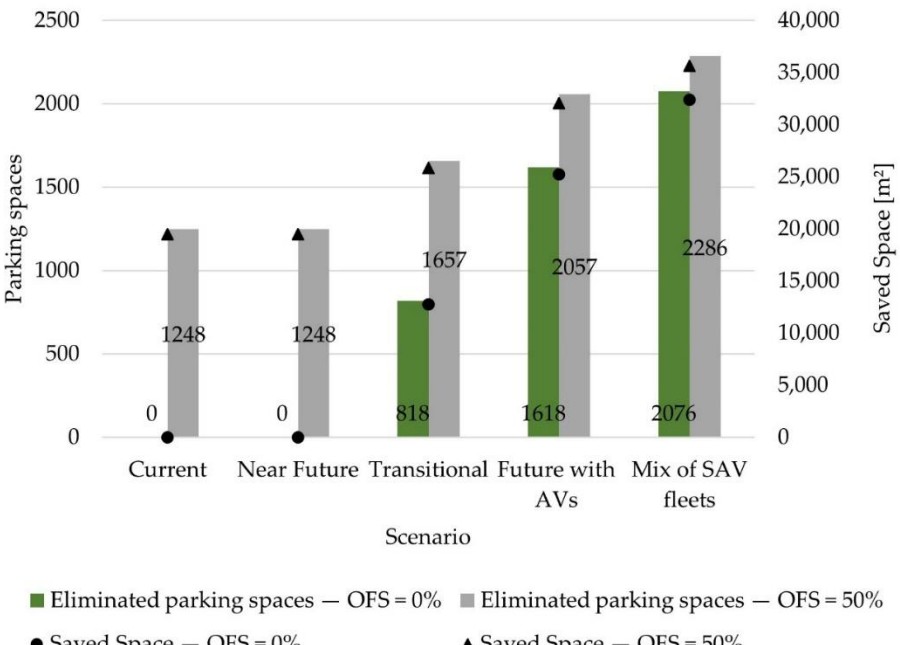

**Figure 3.** Saved Space generated by AVs use.

Figure 4 illustrates the environmental savings in air pollution (according to Equation (8)) and energy use (according to Equation (9)). The figure presents the percentage of SAV fleet compared to conventional vehicles and the summarised parking environmental savings. In this research, parking environmental savings are considered the number of parking spaces saved with the use of AVs. In other words, AV use decreases the demand for parking spaces, eliminating their environmental impact on materials extraction, transport, and construction. Enhancing SAV fleet produces significant gains for air pollution reduction. Summing the percentages of pollutants does not result in 100% as it is not the overall reduction in air pollution, but a reduction of quantity for each pollutant. Also, more pollutants may contribute to air pollution in reality and they were not considered in this study. Considerations during the calculation were:

- Number of trips/distance would remain the same as recently, and
- The pollution for vehicle running is the same as recently (not considering electric propulsion).

Figure 5 shows the quantity of pollutants that green space with shrubs might remove if the space saved is repurposed for green spaces. The result of repurposing 43% of urban space with green areas (result from the questionnaire) is proportional.

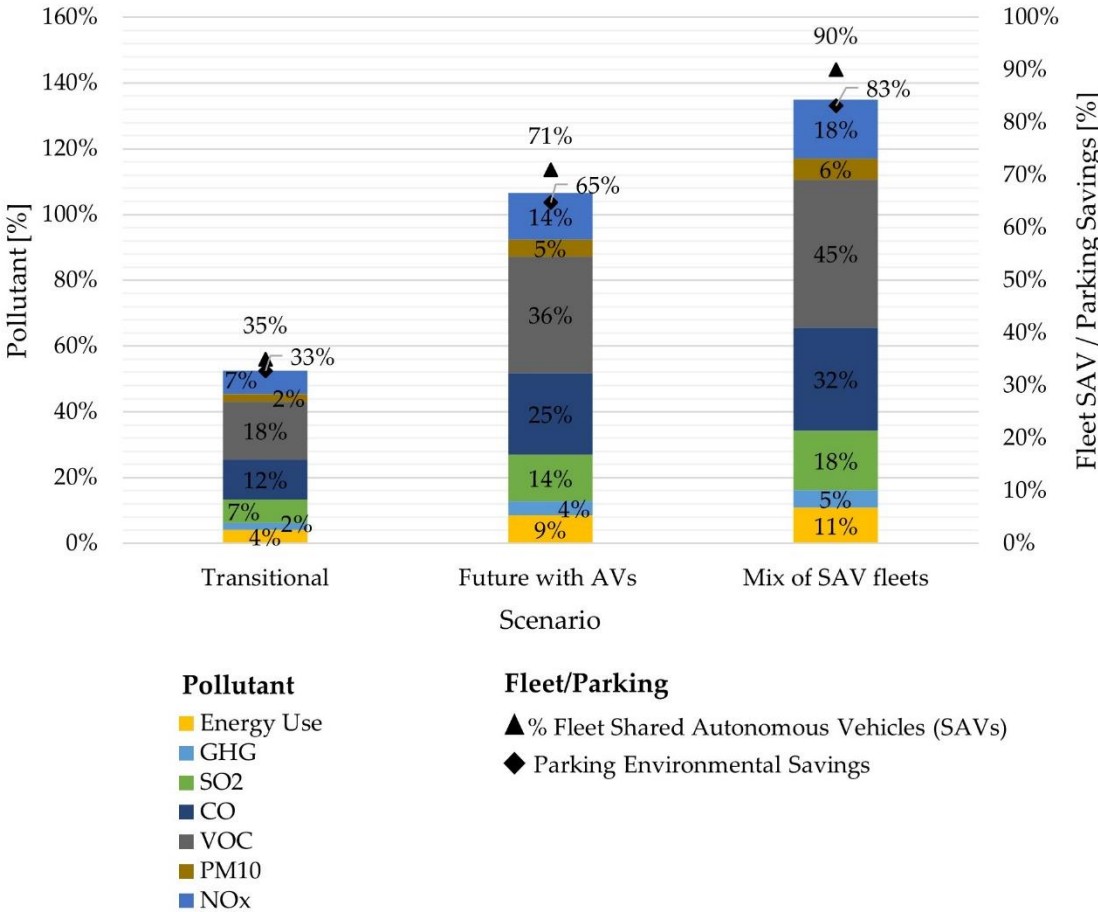

**Figure 4.** Environmental savings—SAV use.

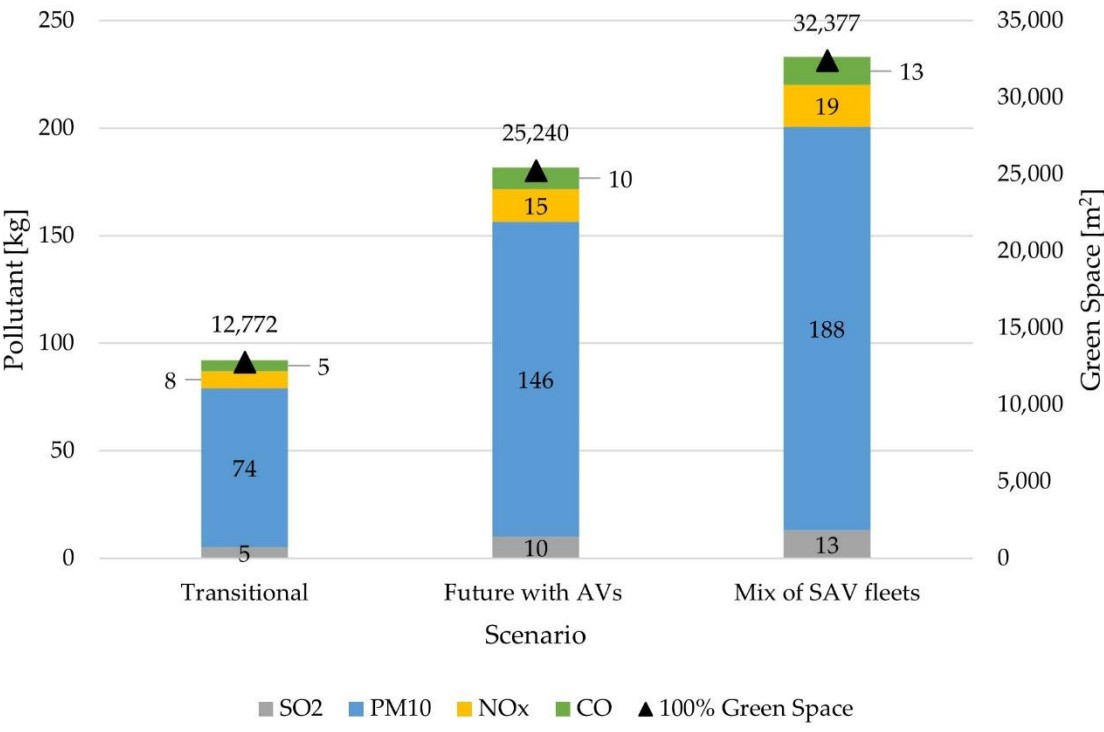

**Figure 5.** Environmental Gains—100% Green Space.

## 5. Discussion

As Scenarios 1 (Current) and 2 (Near Future) are based on conventional vehicles and fleet, the parking spaces needed are constant. Parking spaces cannot be eliminated. Furthermore, Scenario 2 presents a significant increase in private ownership (questionnaire result), which increases parking demand. This result conflicts with respondents' interests, who desire a more liveable city, prioritising green spaces and active transport. Thus, citizens need to understand how their choices regarding shared ownership adoption influence liveability.

Scenario 3 (Transitional) presents half of the fleet as AVs (52%). However, 70% of the total fleet is private vehicles (48% conventional private, 10% private AV, 7% PSAV), showing that only 818 on-street parking spaces can be eliminated (30% of the total of parking spaces in the study area). Thus, the adoption of SAVs has more impact on the reduction of parking spaces than only using AVs privately.

On the other hand, Scenario 4 (Future with AVs) has around 70% of the fleet being used by shared systems, double the proportion of Scenario 3. As a result, 1618 parking spaces can be saved for other use (around double what it saved in Scenario 3), which indicates that the space saved increases in the same proportion as the proportion of sharing. With more SAV, the proportion of saved space provided by off-street parking decreases compared to on-street parking. Also, Scenario 5 (Mix of SAV fleets) confirms this claim.

Generally, off-street parking seems an effective solution for providing space saved if shared systems are not accepted. It is also applicable when the preference for on-street parking is high, but the availability is low due to limited urban space. As off-street parking usually has a higher cost, subsidies might be applied to reduce its price and/or parking policies that might restrict the number of on-street parking spaces, improving the acceptance of off-street parking.

In addition to the findings in Scenario 3, the comparison between Scenario 5 (Mix of fleets) and Scenario 4 (Future with AVs) testifies that car ownership is one of the most critical aspects for urban space transformation as it determines parking demand. Therefore, the adoption of SAV and higher occupancy represent enormous gains for the urban space repurposing. Besides, when SAVs are well-adopted, building off-street parking spaces will serve with value-added services such as charging, maintenance, cleaning, and waiting areas for new users, which may increase the acceptance of off-street parking over the on-street option. Then, on-street parking may serve multifunctional purposes that vary according to the local needs and time of the day (e.g., loading/unloading activities in the early morning, spaces for restaurants at lunchtime, parking for charging electric vehicles at night, etc.).

Construction and maintenance of parking infrastructure have a significant contribution to air pollution too. The exchange from on-street parking to off-street parking does not mitigate air pollution, and it is necessary to reduce parking demand. As SAV use may cause this reduction, the decrease in emissions compared to conventional vehicles is an additional advantage. Positive impacts on air quality caused by SAV use may be enhanced if it is combined with on-demand mobility service and electric propulsion.

Green spaces may considerably extend the environmental benefits to air pollution. Taking the scenario with the lowest AV fleet (Scenario 3: 52% AV fleet) as an example and considering repurposing with 100% of green space, the air pollution reduction in one year overcomes the one produced by parking infrastructure, except for CO (3.6 kg of $SO_2$, 5.2 kg of CO, 6.4 kg of $NO_x$, and 27 kg of $PM_{10}$).

Splitting proportions among the priorities may be more beneficial than using the total space saved for only one purpose. Consequently, it may attend to the needs of more individuals and amplify the gains to the environment.

Note that different results are found depending on the city to which the study is applied because of the settlement structure, trip durations, distance, and residents' needs. However, the consequences generated by enhancing the SAVs use regarding the gains in urban space through reducing parking demand are clearly pronounced.

## 6. Conclusions

The transformation of urban space as a consequence of AVs use has to be predicted in advance in order to anticipate the planning of changes in the urban environment. As the estimation has several uncertainties, building scenarios that assess future periods are needed to understand the alterations better. The main contribution of this study is a scenario building method considering parking demand through fleet size, modal share, car ownership, as well as the preference of parking and urban space repurposing.

The key findings of the research are:

- Five scenarios can depict the possible future situations according to the literature review and the results of a questionnaire survey.
- Car ownership is the most critical aspect of transformation in urban space due to its potential to decrease parking demand.
- SAVs can significantly minimise air pollution caused by parking infrastructure as they decrease parking demand (from 33% up to 83%). Also, for other pollutants, the reduction can reach 45%.
- SAVs use positively impacts the amount of space for repurposing (around 83%) through reducing parking demand.

For future research, we aim to involve more aspects in the scenario building method, such as changes in population, location of households, on-street parking for loading, and charging. Furthermore, our focus is on the investigation of dedicated parts. For example, focusing on the modal share will allow us a better investigation of the effects of micro-mobility and public transport on parking demand and urban space transformation. Further aims are to incorporate pre-available data to validate the method, refine scenarios with a larger sample size, and a multi-country questionnaire. Moreover, the impact of innovative on-demand mobility services will be determined in a systematic approach.

**Author Contributions:** Conceptualization, D.S., D.F. and C.C.; methodology, D.S.; investigation, D.S. and D.F.; writing—original draft preparation, D.S. and D.F.; writing—review and editing, D.S., D.F. and C.C.; visualization, D.S. and D.F.; supervision, D.F. and C.C.; project administration, D.F. and C.C.; funding acquisition, C.C. All authors have read and agreed to the published version of the manuscript.

**Funding:** The research reported in this paper and carried out at the Budapest University of Technology and Economics has been supported by the National Research Development and Innovation Fund (TKP2020 Institution Excellence Subprogram, Grant No. BME-IE-MIFM) based on the charter of bolster issued by the National Research Development and Innovation Office under the auspices of the Ministry for Innovation and Technology. EFOP-3.6.3-VEKOP-16-2017-00001: Talent management in autonomous vehicle control technologies. The Project is supported by the Hungarian Government and co-financed by the European Social Fund. The research was supported by the Ministry of Innovation and Technology NRDI Office within the framework of the Autonomous Systems National Laboratory Program.

**Institutional Review Board Statement:** Not applicable.

**Informed Consent Statement:** Not applicable.

**Data Availability Statement:** Data are contained within the article.

**Conflicts of Interest:** The authors declare no conflict of interest. The funders had no role in the design of the study; in the collection, analyses, or interpretation of data; in the writing of the manuscript, or in the decision to publish the results.

## Appendix A

The questionnaire survey is presented below.

A. General Travel Behaviour

    1. Mark the sentence which best describes you (Evaluation: Current Car ownership):

        a. I have my own car (Private)

      b.      I share the vehicle with members of the family (Private shared)

      c.      I drive a shared car (Carsharing with single occupancy)

      d.      I take a taxi (Sharing with single occupancy)

      e.      I share the same vehicle with other people (Ridesharing/Carsharing with multiple occupancy)

2. Choose the three most used (please consider the number of trips) modes by you (Evaluation: Current modal share):

      a.      Public Transport

      b.      Bicycle

      c.      Walking

      d.      Car as a driver

      e.      Car as a passenger

      f.      Taxi

      g.      Motorbike

      h.      Carsharing

      i.      Bike-sharing

      j.      Ridesharing as a passenger

      k.      Ridesharing as a driver

B. Parking

3. Choose the two most used (please consider the number of parking sessions) parking facility types by you (Evaluation: identifying the most used type of parking, confirming whether on-street is the most used):

      a.      Private parking at home (e.g., garage)

      b.      Private parking (e.g., a paid building garage which has a private owner)

      c.      Workplace

      d.      Public parking at curb-side

      e.      Supermarkets, markets, shops

      f.      Parking and Ride (P + R) parking facility

C. Willingness to change

4. What is the probability of you using an autonomous vehicle? (Evaluation: user acceptability):

      a.      None

      b.      Very unlikely

      c.      Unlikely

      d.      Likely

      e.      Very likely

5. Are you willing to change your current car ownership? Car ownership: you have a car, you share a car with member of your family, you use carsharing or ridesharing (Evaluation: sharing acceptability):

      a.      No, the current option will remain.

      b.      Yes, I would buy my own car (private ownership).

      c.      Yes, I would share my car with members of my family in order to reduce the number of cars that we have (private shared ownership).

      d.      Yes, I would use taxi or carsharing as a driver (shared with single occupancy ownership).

      e.      Yes, I would use ridesharing as a driver (shared with multiple occupancy).

      f.      Yes, I would use ridesharing as a passenger (shared with multiple occupancy).

6. Mark the options which describe you best when completing the sentence: I would use AVs . . . (Evaluation: Situations in which respondents would use AVs, adapted [57]):

      a.      To avoid parking

  b.  When personal vehicle is unavailable (maintenance or repairs)
  c.  As an alternative to driving (e.g., after drinking alcohol)
  d.  For long trips
  e.  For short trips
  f.  Other
  g.  Never

7. Mark the options which describe you best when completing the sentence: I would share an AV with someone at the same time . . . (Evaluation: Situations in which respondents would use SAVs with multiple occupancy, adapted [57]):

  a.  When riding alone
  b.  When riding with an adult family member or friend
  c.  When riding with my child
  d.  Only at times of day I think are safe
  e.  For work
  f.  For shopping
  g.  For recreational trips
  h.  For all trips
  i.  I would not use the service

D. Priorities of Citizens/Decision Makers

8. If parking spaces on streets could be eliminated, how would you use this additional space? Order according to priority, 1 is the highest priority and 9 is the lowest priority (Evaluation: identifying the priorities of decision makers regarding changes in urban space):

  a.  New buildings
  b.  Additional traffic lane
  c.  Green spaces (e.g., gardens and parklets)
  d.  Pick-up and drop-off points in every block
  e.  Bike lanes
  f.  Multifunctional zones (loading/unloading, food trucks, parklets)

E. Socioeconomic Data

9. Age

  a.  18–35
  b.  36–60
  c.  >60

10. Gender

  a.  Female
  b.  Male
  c.  Other

11. Do you have a driving licence?

  a.  No
  b.  Yes

12. Employment

  a.  Student
  b.  Self-employed
  c.  Employee in a private company
  d.  Employee in the public sector
  e.  Retired

13. Adults in the household

  a.  1
  b.  2
  c.  >2

14. Children in the household

    a. 0
    b. 1
    c. 2
    d. 3
    e. >3

15. Education level

    a. Studied until high school
    b. Technical College
    c. Bachelor's degree
    d. Master's degree
    e. Doctoral degree

16. Income level per person (HUF) per month

    a. <=100,000
    b. 100,001–200,000
    c. 200,001–300,000
    d. 300,001–400,000
    e. 400,001–500,000
    f. >500,000

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
