# Peer review of "Autonomous Vehicle Use and Urban Space Transformation: A Scenario Building and Analysing Method"

_sustainability, doi:10.3390/su13063008_

Round 1
Reviewer 1 Report
The article focuses on ownership, acceptability of AVs and shared use, parking preferences, and urban space repurposing during the elaboration of a novel method. The article presents some promising results , I have the following suggestion to improve the article:
Please ensure that when you upload a manuscript please remove th markup of the document as its very difficult to read
Please insert references to all the equations that are not original.
The quality of the article can be further solidified by adding recent and relevant references. I suggest adding more references to enrich the article. For example:
The author states "These phenomena reduce the amount of space needed in parking areas. Decreasing the demand for parking and fleet size would
considerably eliminate part of these emissions" I suggest strengthening the sentence by adding relevant recent studies as references like:
- (2019). Design, Economics, and Real-Time Optimization of a Solar/Natural Gas Hybrid Power Plant (Doctoral dissertation, The University of Utah).
Reviewer 2 Report
The authors made some suitable changes of the text. It is clearer than in the previous version on the paper. The authors mentioned the role of urban public transport. They changed the role of a survey involving relative small number of 150 respondents to an initial survey. It means that main previous comments are reflected and the paper is improved now.
On the other hand, the topic still seems to me to be too much complex for one paper. I would like to recommend, to consider the level of resolution carefully and to present the results in the way related to this resolution. It can mean to be more general and follow less aspects only or to select a part of research and present this part on such detail level.
Author Response
Please see the attachment.

This manuscript is a resubmission of an earlier submission. The following is a list of the peer review reports and author responses from that submission.
Round 1
Reviewer 1 Report
Line 219 – Check reference link.
Line 256 – Figure 1 – I would like to recommend to insert one new element into this scheme. This element can be for conditions within SAVSO (or other AV) system. For example, when there will be small number of vehicles within such system, waiting times on coming of shared vehicle can be longer and the people will possibly prefer their own cars (fleet size of all vehicles could be increased). Similar effects can be related to price or to other operation conditions (e.g. if it will be impossible to leave the city by such car etc.). Finally, these factors can have also impact on demand after parking spaces etc. in my opinion.
Line 350 – I am not sure, if the statistical sample consisted of 150 respondents is suitable for modelling of impacts of such strategic decisions in the city of Budapest.
Table 4 – Use dot in spite of comma in decimal numbers in English text.
There is a relative high volume of specific numbers applied in computation. On the other hand, in my opinion, some other crucial data needed for such complex modelling are not mentioned. For example, what will be the role of urban public transport system in your scenarios? For example, AVs can possibly attract former passengers not only due to higher comfort of driving, but also due to possible reduction of traffic flow intensity (reduction of congestion).
I would like to recommend to divide the paper into more independent papers, each dedicated to one part of this complex methodology. It can be more illustrative for the readers and you can better express data collection and processing as well as overall impact on the final state. You can also better abstract from effects with unknown impacts (like e.g. mentioned role of public transport).
Reviewer 2 Report
In this article scenarios with significant shared AV use show that parking demand may be minimised and urban space repurposing has the highest potential; furthermore, AV use and sharing acceptance may decrease the fleet size and alter the type of shared mode (multiple occupancies). The developed scenario building method serves as a base for future studies. The produced scenarios allow the researchers to focus on the analysis of the impacts caused.
I suggest the author adding the clarification regarding the novelty of the work. I suggest adding a paragraph on the novelty.
I suggest checking the article thoroughly for grammatical mistakes and formatting errors before submitting to the journal. For example there are at least three places in the article where the message "Error! Reference source not found." shows up. These kind of mistakes is a sign of negligence for the author.
I suggest adding references to all the equations that are not original.
I also suggest adding a bit more on the environmental benefits of AV and also mention recent environmental studies as reference for example:
- Design, Economics, and Real-Time Optimization of a Solar/Natural Gas Hybrid Power Plant (Doctoral dissertation, The University of Utah).
- (2016). The environmental impact of autonomous vehicles depends on adoption patterns.
Reviewer 3 Report
Dear authors,
thanks for submitting this article. While I think the topic and approach of the research are surely very interesting for the readers, I do not think the paper is ready for publication yet. There a number of major issues that the authors should address:
- In your literature review you neglect almost completely the literature on parking policy and management, that seems to me very important for your scenario building
- In your scenario building method (summarised in figure 1) there are many concepts/relations that are questionable:
- You state that fleet size (I guess you mean car ownership) has a unilateral influence on parking preference; why is that?
- type of ownership and parking preference should have a bilateral influence;
- Because you miss the literature on parking management, you miss parking options/availability in your scheme. From literature we know that the availability of parking has a strong influence on car ownership (fleet size) and car use (Modal Share).
- Why should parking demand have a unilateral influence on urban space repurposing priorities? It might have an influence on the quantity of urban space that can be repurposed, but not on the priority. The priority is based on the political agenda of the city.
- Data sources explained in figure 2: Fleet size should also be based on statistical data
- The survey object of this paper is not provided in the appendix; accordingly, it is not possible to assess all assumptions you have made in your model. Next, there are no descriptives statistics; this does not help either.
- Did you survey citizens or decision makers or both? I think you only surveyed citizens, but you claim you have data about the priorities of decision makers. How did you get this data?
Other minor issues:
- Line 65: why should the possible solutions for urban space repurposing have an influence the scenario's? Free urban space that can be used for other purposes than parking is an outcome of the different scenario's not an aspect of the scenario self. Or do I see it wrongly?
- Line 147: what is pay and park measure?
- Line 148-149: this might come out of a computer simulation, but the 1 km radius rule simply does not exist in the real world.
- Line 159-167: this part of your literature review has little to do with your research.
- Overall in the text: you always use the term of fleet size but I guess you refer to car ownership, right?
- Line 233: there is not really evidence for that in the literature; it's not true that off-street parking is preferred over on-street parking. And this is one of the assumptions of your model.
- Line 292: explain this concept better: why should the type of ownership influence parking demand? Parking demand is a derived demand, influenced by many factors.
- 150 respondents are not that much....
- Line 363-370: personally, I think sound scientific research should not be based only on Deloitte reports...
- Line 413-427: it's not clear in the text how you get this results
- Line 471 (statement in bold): we already know that, this is taking place in many EU cities since years;
Round 2
Reviewer 1 Report
The manuscript has been extended and clarified by the authors. I hope that made changes are helpful and I thank to the authors for it.
It is better to highlight the methodology than the case study (as it is done now), because it is too complex topic to be presented by one paper with all presupmtions and conditions applied.
I would like to accept the manuscript now.
Reviewer 2 Report
After going through the revised version I think the author hasn't put enough time into the comments given. I suggest this article to be further written carefully and descriptive statistics is definitely missing in this article to support thr results.
Reviewer 3 Report
Dear authors,
I think you rushed too much to prepare a new version of the article, the result being an article that is more complex to read and to understand compare to the first version. You should never rush scientific research!
Some issues concerning this second version:
- I think the questionnaire you used has serious flaws in terms of the questions you asked. For example: you cannot ask the same questions to policy makers and citizens; citizens are not able to answer section D, while policy maker might. Additionally, how do you know the difference between a policy maker and a normal citizen? I suppose the survey was anonymous.... Question 5 should have been divided in two questions: one concerning parking at home, one concerning parking at destination. Individual preferences for these two types of parking are usually quite different.
- By not showing the descriptive statistics you make it impossible for the reader to judge the quality of your responses. This especially because you say that you make use of the survey to confirm the findings of the literature. 150 is very low and if I don't even know how many of them are car drivers, how can I trust your results?
- there are still many parts/assumptions in your scenario building that are not clear:
- Line 319: if Als shift demand from public transport, then they don't lead to any reduction in parking demand;
- Line 327: "Operators will be able to choose...": who are these operators? AV's operators?
- Figure 1 is not coherent with the description above; second quadrant in the grey square: only car ownership or car ownership and sharing type?
- Line 173: a long-term measure is different from long-term parking
- Line 48-51: in your new introduction you state that the combination of primary and secondary sources is the novelty aspect in your methodology. Is that so? I am not an expert in scenario building, but such a combination of sources is very normal in research.